# Infection Prevalence of Microsporidia *Vairimorpha* (*Nosema*) spp. in Japanese Bumblebees

**DOI:** 10.3390/insects14040340

**Published:** 2023-03-30

**Authors:** Takahiro Yanagisawa, Yuto Kato, Maki N. Inoue

**Affiliations:** Department of Agriculture, Tokyo University of Agriculture and Technology, 3-5-8 Saiwai-cho, Fuchu, Tokyo 183-8509, Japan

**Keywords:** alien species, *Bombus terrestris*, *Vairimorpha bombi*, spore density, *Vairimorpha* sp.

## Abstract

**Simple Summary:**

Microsporidia are intracellular parasites that have detrimental effects on their various invertebrate and vertebrate hosts. The microsporidia pathogenic to honeybees and bumblebees belong to the genus *Vairimorpha* (previously *Nosema*). Among these, *V. bombi* is the most widespread species worldwide and it infects a variety of bumblebee species. In this study, we investigated the infection of *V. bombi* among Japanese native bumblebees and the alien species, *Bombus terrestris*, established in Japan, using PCR and microscopy. We discovered that *V. bombi* infected the subgenus *Bombus* s. str. species/subspecies with a low prevalence in areas both with and without the presence of *B. terrestris*, suggesting that *V. bombi* may be native to Japan. In contrast, a newly discovered microsporidian parasite that is phylogenetically distinct from *V. bombi* was found to infect the subgenus *Diversobombus* species/subspecies with a high prevalence. Hence, if the new microsporidian species does not negatively affect the host and is vertically transmitted from queen to offspring, it may coexist with its host. In addition, this may suggest that *V. bombi* and the new microsporidium have different life history strategies.

**Abstract:**

Microsporidia are spore-forming intracellular parasites of various invertebrates and vertebrates. *Vairimorpha bombi* negatively affects the fitness of bumblebees and its prevalence correlates with declining bumblebee populations. The invasive alien species *Bombus terrestris* colonized Japan and possibly introduced new parasites. To assess the infection prevalence of *V. bombi* in Japanese bumblebees and *B. terrestris*, we investigated *V. bombi* infections using PCR and microscopy. The prevalence of sporulating *V. bombi* infections in three *Bombus* s. str. species/subspecies was low, whereas that of non/low-sporulating *Vairimorpha* sp. infections in three *Diversobombus* species/subspecies was high. Invasive *B. terrestris* showed low prevalence of non/low-sporulating *V. bombi* infections and shared the same *V. bombi* haplotype with *B. hypocrita* found in Hokkaido, where *B. terrestris* is present, and in Honshu, where *B. terrestris* is absent. Although *V. bombi* may have been introduced with *B. terrestris* colonies imported from Europe, it seems to be originally distributed in Japan. Furthermore, a new *Vairimorpha* sp. was found in Japanese bumblebee species. *V. bombi* and *Vairimorpha* sp. showed different organ and host specificities in bumblebees. There are no reports on the specific effects of other *Vairimorpha* spp. on bumblebees; further studies are needed to clarify the individual characteristics of *Vairimorpha* spp.

## 1. Introduction

Bumblebees (Apidae: Hymenoptera) are eusocial insects with annual colony cycles. Queens emerge from hibernation in spring and establish new colonies, each producing hundreds of workers during the summer. Declines in wild bumblebee populations have been reported mainly in Europe and the United States; four bumblebee species in Europe have disappeared from eleven countries in the last sixty years [1], and the prevalence of microsporidia was high in bumblebee species with declining populations in the United States [2,3].

Microsporidia are intracellular spore-forming parasites of a variety of invertebrates and vertebrates and sometimes cause extensive damage and death of the hosts [4]. For example, *Nosema bombycis*, a parasite of silkworms, kills most orally inoculated larvae [5]. *Nosema thompsoni*, which infects the harlequin ladybird *Harmonia axyridis*, was found to successfully reduce the survival rate of the European native ladybird *Coccinella septempunctata* after experimental inoculation [6]. *Vairimorpha* (*Nosema*) *bombi*, redefined by Tokarev et al. [7], is a parasite of bumblebees, first discovered by Fantham and Porter in 1914 [8], and is considered to have spread worldwide with commercialized bumblebee colonies [3]. Infection with *V. bombi* negatively affects the fitness of bumblebees, shortening adult longevity and reducing colony size [9,10]. During the past decade, surveys of *V. bombi* infections among bumblebees have been conducted worldwide; their prevalence varied, ranging from 1.2% to 66.5% [2,11,12,13,14,15,16,17,18,19,20,21,22,23,24]. In these studies, light microscopy and/or PCR were used to diagnose infections, but microsporidia infections were often undetectable using microscopy. Rutrecht et al. [25] reported 25% more *V. bombi* infections using PCR diagnosis than with microscopy, and Blaker et al. [11] found high levels of PCR-positive infections with no detectable spores in bumblebees. Thus, a combination of PCR and microscopy is important for surveying the infection prevalence and status of *V. bombi* in wild bumblebees.

The European bumblebee *Bombus terrestris*, introduced to different countries for greenhouse crop pollination, is considered to be one of the factors leading to reduced populations of native pollinators and the species has now been established in New Zealand [26,27,28], Israel [29], Chile [30,31], Argentina [32,33], Australia [34,35], and Japan [36]. In northern Japan, *B. terrestris* has populated wide areas of Hokkaido [37,38,39] since its introduction in 1991 [40]. In open agricultural areas in southern Hokkaido, a drastic decrease in the native consubgeneric species *B. hypocrita sapporoensis* was associated with a rapid increase in *B. terrestris* measured over 3 years from 2003 to 2005, suggesting a competitive replacement of the native by the invading species [36]. The early emergence of its queens in the spring [36], high reproductive ability [41], and superior foraging ability [31,42] could be reasons for *B. terrestris*’ dominance and could lead to further expansion of this species.

However, few studies exist overall on the parasites introduced to bumblebees, and it remains unclear whether the introduction of new parasites affects native bumblebee species. One previous study showed that *V. bombi*-like spores were found in commercial colonies of *B. terrestris* imported from the Netherlands and experimentally transmitted them to the native bumblebees *Bombus hypocrita hypocrita* and *B. diversus diversus* [43]. Furthermore, *V. bombi* detected in *B. terrestris* collected in Hokkaido had the same nucleotide sequence as *V. bombi* from bumblebees in Britain [44]. In its native ranges, *B. terrestris* was infected by *V. bombi* with a relatively high prevalence at approximately 25%, sometimes more than 50% [13,14,24]. Since introduced for pollination in Chile, *B. terrestris* also has been established in Argentina’s Patagonia region [45] and was highly infected by *V. bombi* (37%), suggesting a potential threat to Argentina’s native bumblebees [46].

In this study, we aimed to determine the infection prevalence and status of *V. bombi* in Japanese bumblebees and *B. terrestris* collected from Hokkaido, where *B. terrestris* is established, and Central Japan, where *B. terrestris* has not established itself. We examined the infected organs, assessed the density of microsporidia, and demonstrated the phylogenetic relationship of the reported *V. bombi* isolates.

## 2. Materials and Methods

### 2.1. Sampling of Bumblebees

To investigate the prevalence of *Vairimorpha* spp. among bumblebee species, bumblebee sampling was conducted in Hokkaido, where *B. terrestris* is established, and in Central Japan, where *B. terrestris* is not established (Appendix A). Specimens were collected in Biei and Kamifurano, Central Hokkaido in 2014, 2016, and 2018; Hokuto, Yamanashi and Minamiaiki, Nagano (west side of the Okuchichibu mountain area, west OMA) in 2014 and 2016; Koshu, Yamanashi (east side of the Okuchichibu mountain area, east OMA) in 2016, 2019, and 2020; and in Hinohara, Tokyo in 2016. Additional collections were conducted in Oshino, Yamanashi, and Hara, Nagano in 2016. The midgut of each specimen was removed and crushed in 200 μL 1× PBS (1370 mM NaCl, 27 mM KCl, 81 mM Na_2_HPO_4_ 12H_2_O, and 14.7 mM KH_2_PO_4_; pH 7.4), and homogenates were stored at −35 °C for the following experiments.

### 2.2. Detection of Vairimorpha spp. Spores

Using 1 µL of the homogenates spread on a glass slide, spore-forming (herein, “sporulating”, according to Blaker et al. [11]) *Vairimorpha* spp. were examined across the center of the cover glass from the left edge to the right edge under a phase-contrast microscope (Nikon, Tokyo, Japan) (400× magnification).

### 2.3. PCR and Genotyping of Vairimorpha spp. Infections

A 50 μL sample of the midgut homogenates was mixed with 250 μL DNAzol (Molecular Research Center, Cincinnati, OH, USA) and 250 mg glass beads (SIGMA, Kanagawa, Japan) and vortexed at 3000 rpm for 5 min (CM-70M-07, ELMI, Riga, Latvia). Then, 250 μL DNAzol and 1 μL proteinase K (20 mg/mL) (Merck Bioscience, Darmstadt, Germany) were added prior to incubation for 1 h at 50 °C and centrifugation at 4 °C, 15,400× *g* for 10 min. The aqueous layer was transferred to a new tube, 250 μL of 99% ethanol (FUJIFILM Wako Pure Chemical Corporation, Osaka, Japan) was added, and the mixture was centrifuged at 4 °C, 20,400× *g* for 10 min. The DNA was washed with 70% ethanol, dried for 10 min, resuspended in 20 μL ultrapure water (UPW), and stored at −35 °C.

The PCR assay for *Vairimorpha* spp. detection was performed using 1.0 μL of 10 × Ex *Taq* Buffer (TaKaRa, Shiga, Japan), 0.8 μL of 2.5 mM each dNTP mixture (TaKaRa), 0.2 μL of each forward and reverse primer (10 μM) for small subunit (SSU) rRNA, Nbombi-SSU-Jfl, and Nbombi-SSU-Jrl [47] (Appendix A), 0.05 μL of 5 U/μL Ex *Taq* HS (TaKaRa), 1 μL of template DNA, and 6.75 μL of sterilized distilled water (SDW). The PCR profile for the detection of *Vairimorpha* spp. comprised an initial denaturation step at 94 °C for 3 min, followed by 40 cycles at 94 °C for 30 s, 50 °C for 30 s, and 72 °C for 30 s, and a final extension at 72 °C for 5 min. DNA presence was confirmed by PCR using β-actin primers [48] (Appendix A) at an annealing temperature of 55 °C. The PCR products were purified using a QIAquick PCR Purification Kit (Qiagen, Hilden, Germany), and sequencing reactions were performed using the BigDye Terminator Kit ver. 3.1 (Applied Biosystems, Foster City, CA, USA). PCR products were directly sequenced using a 3700 DNA analyzer (Applied Biosystems, Foster City, CA, USA). The cloned plasmid DNA was used as a positive control in all PCR assays: *Vairimorpha* spp.-like spores were harvested from a male *B. hypocrita sapporoensis*, which was collected in Hokkaido in 2014, and spore DNA was subjected to SSU rRNA gene amplification using the primers Nbombi-SSU-Jfl and Nbombi-SSU-Jrl; the PCR products were purified. The purified PCR products were ligated with the T-vector pMD20 (TaKaRa) using a ligation mix (TaKaRa) as per the manufacturer’s instructions. Competent cells (*Escherichia coli* JM109; TaKaRa) were transformed with plasmids. Plasmid DNA was extracted using the QIAprep^®^ Spin Miniprep Kit (Qiagen) (Hilden, Germany) according to the manufacturer’s protocol and stored at −25 °C.

A phylogenetic tree was constructed based on the SSUr RNA gene sequences amplified from *Vairimorpha* spp. using the primers SSUrRNA-f1 and SSUrRNA-r1c [17] (Appendix A) and gene sequences obtained from GenBank using MEGA 7.0 software [49]. After manual sequence editing, they were aligned and the maximum parsimony method was used to construct a phylogenetic tree for clustering haplotypes. To test the reliability of each clade in the tree, 1000 bootstrap resamplings were performed.

### 2.4. Organs Infected with Vairimorpha spp.

Four organs (midgut, head, thorax, and legs) of bumblebees infected with *Vairimorpha* spp. were examined for signs of infection using microscopy and PCR to demonstrate its systemic infectivity. Two individuals from each species were used in subsequent experiments. To eliminate spores from the body surface of bumblebees, they were washed with 1× PBS. Then, their head, thorax, and legs were separated using scissors sterilized by autoclaving; only the femur was used for the “legs” sample after separation from other parts. For *B. diversus diversus*, an ovary was also isolated and checked for *Vairimorpha* spp. infection to examine the possibility of vertical transmission. Each organ from two individuals was homogenized using a sterile mortar with 200–400 µL of DNAzol and pooled. The homogenates were then transferred to 1.5 mL microtubes.

DNAzol was added to the homogenates up to the 1 mL mark, and 500 mg of glass beads was added. DNA was extracted as described in Section 2.3, with 4 µL proteinase K used here. PCR was performed with 30 ± 5 ng of DNA.

### 2.5. Quantitative PCR

To quantify the severity of *Vairimorpha* spp. infection among bumblebees, qPCR was performed using a StepOnePlus Real-Time PCR System (Applied Biosystems, Foster City, CA, USA). The reaction mixture contained 10 ng DNA, 30 μM of each primer, and 5 μL FastStart Universal SYBR Green Master Mix (Roche, Basel, Switzerland). The cycling conditions were as follows: 2 min at 50 °C and 10 min at 95 °C, followed by 40 cycles of 15 s at 95 °C and 1 min at 60 °C. Dissociation curve analysis of the amplified product was performed after amplification using the specific primers BOMBICAR-F and BOMBICAR-R [50] (Appendix A) with the following conditions: 15 s at 95 °C, 20 s at 60 °C, and 15 s at 95 °C. Negative (UPW) controls were used for each run. For each sample, the numbers of *Vairimorpha* spp. gene copies were calculated using the *C_t_* values (mean of two replicates) measured against a standard curve, which was generated using 10-fold serial dilutions from 3.0 × 10^2^ to 3.0 × 10^−3^ DNA extracted from a solution of 9.8 × 10^5^ spores/mL. The spore density, expressed as the number of spores per 100 ng of DNA, was calculated from duplicate measurements per sample.

## 3. Results

### 3.1. Prevalence of Vairimorpha spp.

We examined 903 bumblebee specimens belonging to 12 species and subspecies collected in four regions of Japan (Hokkaido, Tokyo, Nagano, and Yamanashi). The overall prevalence of *Vairimorpha* spp. across all specimens was 19.4%, varying among bumblebee species (Table 1 and Table 2, Figure 1).

Microscopic examination confirmed the presence of *Vairimorpha* spp. spores in two consubgeneric (*Bombus* s. str.) subspecies of *B. hypocrita*: *B. hypocrita sapporoensis* (nine queens, one male, and two workers collected in 2014 and 2018 in Central Hokkaido) and *B. hypocrita hypocrita* (five queens, one worker, and three males collected in 2016 and a total of 15 individuals, the caste were not distinguished, collected in 2019 in east OMA) (Figure 2). Microsporidia spores were not observed in any of the remaining samples.

PCR analysis revealed that *Vairimorpha* spp. exhibited a low prevalence in Central Hokkaido: 0.8%, 1.4%, and 2.4% of *B. terrestris* specimens and 10.3%, 11.9%, and 6.7% of *B. hypocrita sapporoensis* specimens from Hokkaido were positive for *V. bombi* in 2014, 2016, and 2018, respectively. Additionally, one *B. diversus tersatus* worker collected from Central Hokkaido in 2014 was positive for *Vairimorpha* spp. In Central Japan, a low prevalence of *Vairimorpha* spp. was found in two consubgeneric (*Bombus* s. str.) species *B. ignites* and *B. hypocrita hypocrita*: 1.6% of *B. ignitus* and 7.6% of *B. hypocrita hypocrita* collected from west OMA in 2014 and 2016, respectively, were positive for *Vairimorpha* spp. The prevalence of *Vairimorpha* spp. in *B. hypocrita hypocrita* collected from east OMA was relatively high at 60% in 2016 but decreased to 4.3% in 2020. The prevalence of *Vairimorpha* spp. in *B. honshuensis* was low at 10% in specimens from east OMA and 0% in specimens from west OMA. The highest prevalence of *Vairimorpha* spp. was found in two consubgeneric (*Diversobombus*) species, *B. diversus diversus* and *B. ussurensis*; in 2014, 100% of *B. diversus diversus* collected from west OMA were positive, and in 2016, 100%, 100%, 95.6%, 100%, and 77.8% of *B. diversus diversus* collected from Hinohara, Oshino, west and east OMA, and Hara, respectively, were positive. Similarly, 100% of *B. ussurensis* collected from west OMA in 2016 showed positive PCR results for *Vairimorpha* spp.*i*.

### 3.2. Organ Specificity of Vairimorpha spp.

The infection of *Vairimorpha* spp. in four organs of bumblebees (midgut, head, thorax, and legs) was investigated using PCR analysis. Two queens of *B. hypocrita hypocrita* from Yamanashi were sporulating *Vairimorpha* spp. PCR-positive specimens, and two queens of *B. diversus diversus* from Nagano and two workers each of *B. ussurensis* and *B. honshuensis* from Yamanashi were non/low-sporulating *Vairimorpha* spp. PCR-positive specimens.

In the midgut and leg homogenates of two workers of *B. hypocrita hypocrita*, more than five *Vairimorpha* spp. spores (++) were observed under a phase-contrast microscope at 400× magnification (Figure 3A), whereas 1–4 spores (+) were observed in the head and thorax homogenates (Figure 3A). *Vairimorpha* spp. spores were not observed in any of the *B. diversus*, *B. ussurensis,* or *B. honshuensis* specimens.

Using DNA extracted from *B. hypocrita hypocrita*, PCR analysis revealed amplified *Vairimorpha* spp.fragments of approximately 300 bp from all four organs (Figure 3B). For *B. honshuensis*, *B. diversus diversus*, and *B. ussurensis*, similar fragments were detected in the midguts, thoraxes, and legs, but not in the heads (Figure 3B). DNA was also extracted from the ovaries of *B. diversus diversus*, and an amplified *Vairimorpha* spp. fragment of 300 bp was detected.

### 3.3. Spore Numbers of Vairimorpha spp. among Bumblebees

The number of *Vairimorpha* spp. spores in the midguts per 100 ng DNA containing the host gene were estimated using quantitative PCR in the following specimens: (1) sporulating *Vairimorpha* spp. PCR-positive bees, (2) non/low-sporulating *Vairimorpha* spp. PCR-positive bees, and (3) non/low-sporulating *Vairimorpha* spp. PCR-negative bees (Appendix A, Figure 4). The highest spore numbers were found in the sporulating *Vairimorpha* spp. PCR-positive bees: approximately 1.6 × 10^5^ to 1.1 × 10^6^ spores/100 ng DNA in bees with more than five spores/per visual field of the microscope, and approximately 2.5 × 10^4^ to 1.4 × 10^5^ spores/100 ng DNA in bees with less than five spores/per visual field of the microscope. Conversely, low spore numbers, approximately 8.9 to 1.2 × 10^3^ spores/100 ng DNA, were found in non-sporulating *Vairimorpha* spp. PCR-positive bees. The detection limit for spore concentration as assessed by microscopy was <10^3^ spores/100 ng DNA. According to Blaker et al. [11], this number is the threshold for *Vairimorpha* spp. to be considered non/low-sporulating. As expected, no spores were detected in *V. bombi* PCR-negative bees, because these bees were not infected with *Vairimorpha* spp.

### 3.4. Phylogenetic Analysis Using Microsporidian SSU rRNA

Sequences of 502 bp were directly amplified from the SSU rRNA genes of 10 individuals of *B. hypocrita hypocrita*, 11 *B. hypocrita sapporoensis*, 1 *B. terrestris*, 16 *B. diversus diversus*, 6 *B. ussurensis*, and 2 *B. honshuensis*. BLAST analysis showed the sequences (GenBank accession number: LC756306—LC756308) from the three *Bombus* s. str. species were the same and more than 99% identical to the SSUr RNA of *V. bombi* from Europe, the United States, China, and Russia. The sequences (GenBank accession number: LC756309—LC756312) from the three *Diversobombus* species and *B. honshuensis* were the same and shared more than 98% similarity with *Nosema* sp. from India and China. Phylogenetic analysis using the maximum parsimony method showed that *Vairimorpha* sp. found in Japan belonged to the same clade as *N. bombi*, whereas *Vairimorpha* sp. was close to the clade of *Nosema* sp. D (Figure 5), suggesting that *Vairimorpha* sp. is endemic to Japanese bumblebees, and is hereafter referred to as *Vairimorpha* sp. J.

## 4. Discussion

This study showed that PCR is a more sensitive and specific method for *V. bombi* detection than microscopy is. The prevalence of sporulating *V. bombi* was low (6.8%) among the three *Bombus* s. str. species/subspecies, *B. hypocrita hypocrita*, *B. hypocrita sapporoensis*, and *B. ignitus*, and the prevalence of non/low-sporulating *Vairimorpha* sp. J was high (96.1%) among the three *Diversobombus* species/subspecies, *B. diversus diversus, B. diversus tersatus*, and *B. ussurensis*. The invasive *B. terrestris* showed low prevalence of non/low-sporulating *V. bombi* infections and shared the same haplotype of *V. bombi* with *B. hypocrita* collected in Central Hokkaido and Honshu, Japan.

Previous studies have reported the introduction of alien parasites with *B. terrestris* commercial colonies [51,52,53,54]. For example, *V. bombi* and other parasitic species have been detected in worker bees and pollen of commercial colonies from Europe [55]. Therefore, it is possible that *V. bombi* could have been introduced to and subsequently spread with bees escaping from greenhouses to the field. Consequently, it could have horizontally infected the native bumblebee species. Takahashi et al. [44] reported that the prevalence of *V. bombi* was 7.5% (*n* = 120) and 0% (*n* = 100) in *B. terrestris* and native *Bombus* spp., respectively, in other areas of Hokkaido and Nagano from 2010 to 2013, suggesting that *V. bombi* may have been brought with *B. terrestris* colonies imported from Europe later. However, considering that *V. bombi* was detected not only in *B. hypocrita sapporoensis* in Hokkaido, where *B. terrestris* is present, but also *B. hypocrita hypocrita* in Honshu, where *B. terrestris* is absent, and that the higher prevalence of *V. bombi* in *B. hypocrita* than *B. terrestris* suggests that *V. bombi* originates from Japan. In the future, it will be necessary to distinguish whether *V. bombi* is introduced or native to clarify the status of the accompanying *V. bombi* infestation using molecular genetic techniques.

The prevalence of *V. bombi* in wild bumblebees is often low. Studies using microscopy reported that the overall prevalence of nine different *Bombus* spp. in the United States was 2.9% (*n* = 9909) with the highest prevalence being 37.2% in declining *B. occidentalis* (*n* = 172) [15]. In Europe, 56.1% of the natural population of *B. terrestris* in Sweden (*n* = 380) [14] and 24.7% of *B. terrestris/lucorum* with 9.0% overall prevalence in 10 different *Bombus* spp. and *Psithrus* spp. (*n* = 446) in Switzerland [13] were infected with *V. bombi*. Conversely, PCR analysis suggested the mean prevalence was 1.2% (*n* = 83) for *B. terrestris* in Spain [21], 6.1% for 27 different *Bombus* spp. (*n* = 1009) in China [16], 12.7% for 22 different *Bombus* spp. (*n* = 595) in Russia [18], 10.3% for five different *Bombus* spp. (*n* = 39) in India [22], and 13.6% for four different *Bombus* spp. (*n* = 280) in Thailand [23]. The 6.8% overall prevalence of *V. bombi* in our study was similar to that reported in other areas of the world. However, a high prevalence of *V. bombi* was found in *B. hypocrita hypocrita* collected from Koshu, Yamanashi from 2016 to 2019, suggesting that the spreading of *V. bombi* occurred at the site and ended in 2020 for unknown reasons.

While *V. bombi* has been reported across a wide range of the Northern Hemisphere [3,11,15,56], other *Vairimorpha* (*Nosema*) spp. found in bumblebees have been reported in Asian regions—four species from China and two from Russia [16,17]—indicating a high diversity of microsporidia in bumblebees in Asia. Bumblebees also show high species diversity in Asia, and their geographic origin is considered to be the mountainous region of Asia [57,58]. Based on these facts, the microsporidia of bumblebees may have originated in Asia and then dispersed to Europe and North America as bumblebees expanded their habitats. Specifically, this is a possible explanation for the worldwide occurrence of *V. bombi*.

In this study, a new *Vairimorpha* sp. J was detected in species endemic to Japan, such as *B. diversus*, *B. honshuensis*, and in *B. ussurensis*, which is also distributed in far east Russia and Korea [59]. Organ specificity in bumblebees differed between *V. bombi* and *Vairimorpha* sp. J: *V. bombi* showed systemic infectivity, whereas *Vairimorpha* sp. J did not seem to infect the head tissue of bees (Figure 3). It is possible that spores contaminated on the surface of the body were detected, but considering differences in spore density between these two species, this might be a species-specific difference in infections for target organs. For example, nine *Nosema* species of grasshoppers have organ specificity: *N. locustae* and *N. chorthippi* infect only the fat body, showing host and tissue specificity, whereas *N. pyrgomorphae* infects the midgut, fat body, muscle tissue, and gonad [60]. Alternatively, the difference in infected organs might be related to spore density; the spore density of some specimens with sporulating *V. bombi* was high, whereas that of others with non/low-sporulating *Vairimorpha* sp. J was low. *Vairimorpha* sp. J was highly prevalent in *B. diversus* and *B. ussurensis*. Since *V. bombi* infects multiple subgenera of bumblebees [2,3,11,16,17] and can be artificially inoculated into *B. diversus* [43], it has a wide host range. In contrast, *Vairimorpha* spp. reported in China and Russia also infect various bumblebee subgenera with a low prevalence [16,17]. Of these, *Nosema* sp. D contained microsporidia detected from *B. trifasciatus* (*Diversobombus*), *B. lepides* (*Pyrobombus*), *B. friseanus* (*Melanobombus*), and *B. festivus* (*Festivobombus*), and 83% of *B. trifasciatus* specimens were infected with *Nosema* sp. D, albeit with a small sample size [16]. *Vairimorpha* sp. J was mostly found in long-tongued *Diversobombus* species. These infection patterns, combined with molecular phylogenetic analysis, suggest that *Vairimorpha* sp. J is a distinct species from *V. bombi* and is more closely related to *Nosema* sp. D.

The negative effects of *V. bombi* infection on bumblebees [9,61] and its high prevalence in declining bumblebee species [2] have been reported; however, there are no reports on the effects of other *Vairimorpha* spp. *Vairimorpha* sp. J., which is highly prevalent in *Diversobombus* species/subspecies, but shows low spore density, may be a covert infection, possibly vertically transmitted from queen to offspring. In this scenario, it may provide a fitness benefit to its host by establishing a mutualistic relationship to spread throughout a host population. To clarify the characteristics of *Vairimorpha* spp., it is necessary to conduct a comparative study of *Vairimorpha* spp.-infected and non-infected bumblebees in the future.

In conclusion, we revealed that *V. bombi* and a new microsporidium, *Vairimorpha* sp. J., are infecting the Japanese native bumblebees and discovered that they may have different life history strategies.

## Figures and Tables

**Figure 1 insects-14-00340-f001:**
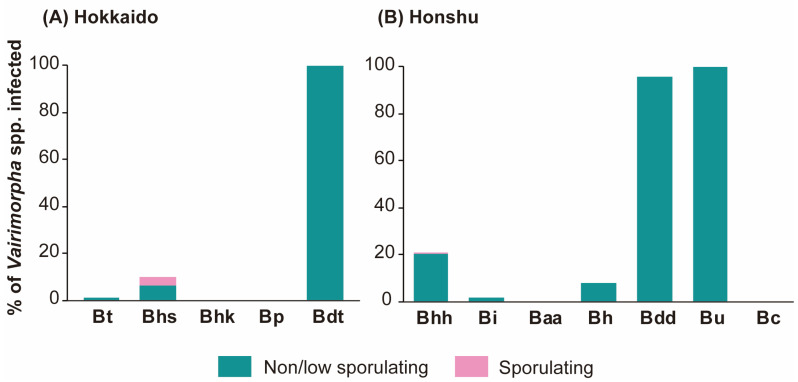
Prevalence of *Vairimorpha* spp. in different bumblebee species. Species names as abbreviations: Bt, *B. terrestris*; Bhs, *B. hypocrita sapporoensis*; Bhk, *B. hypnorum koropokkrus*; Bp, *B. pseudobaicalensis*; Bdt, *B. diversus tersatus*; Bhh, *B. hypocrita hypocrita*; Bi, *B. ignitus*; Baa, *B. ardens ardens*; Bh, *B. honshuensis*; Bdd, *B. diversus diversus*; Bu, *B. ussurensis*; and Bc, *B. consobrinus*.

**Figure 2 insects-14-00340-f002:**
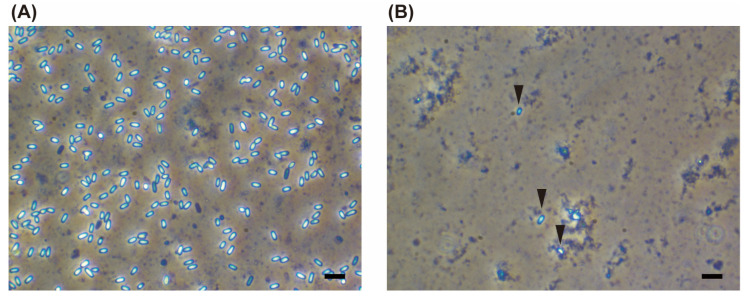
Images of *Vairimorpha* spp.-infected midgut homogenates of *B. hypocrita hypocrita* under a phase contrast microscopy (400×): (**A**) >five spores and (**B**) 1–4 spores. Arrowheads indicate microsporidian spores. Scale bar = 10 μm.

**Figure 3 insects-14-00340-f003:**
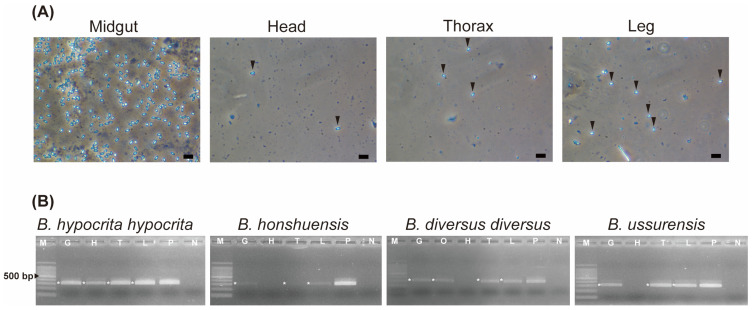
Organ specificity of *Vairimorpha* spp. (**A**) Images of the *Vairimorpha* spp.-infected organs of *B. hypocrita hypocrita* under a phase-contrast microscope (×400): midgut, head, thorax, and leg. Arrowheads indicate microsporidian spores. Scale bar = 10 μm. (**B**) PCR detection of *Vairimorpha* spp. in wild species: *B. hypocrita hypocrita*, *B. honshuensis*, *B. diversus diversus*, and *B. ussurensis*. M: DNA Ladder, G: midgut, O: ovary, H: head, T: thorax, L: leg, P: positive control, N: negative control. Asterisks indicate PCR-positive results.

**Figure 4 insects-14-00340-f004:**
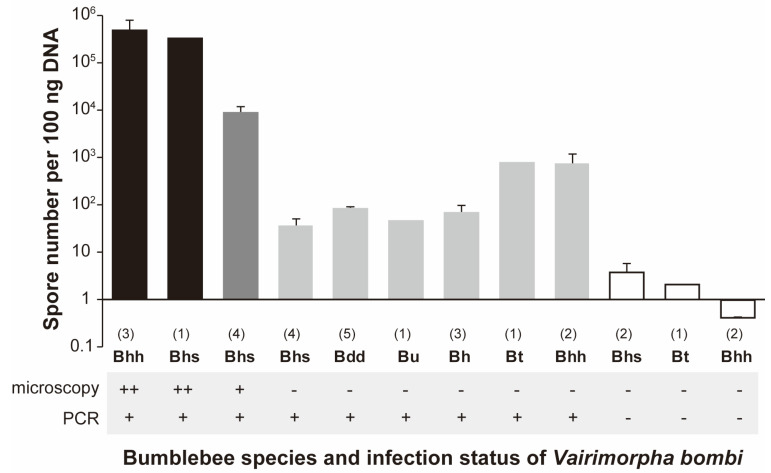
Spore numbers per 100 ng DNA as bar graph quantified using quantitative PCR in each bumblebee species and their *Vairimorpha* spp. infection statuses. Black bar indicates sporulating *Vairimorpha* spp. PCR-positive bumblebees (with more than five spores in a visual field); dark gray bar, sporulating *Vairimorpha* spp. PCR-positive bumblebees (with less than five spores in visual field); light gray bar, non/low-sporulating *Vairimorpha* spp. PCR-positive bumblebees; white bar, *Vairimorpha* spp. PCR-negative bumblebees. Species names as abbreviations: Bhh, *B. hypocrita hypocrita*; Bhs, *B. hypocrita sapporoensis*; Bt, *B. terrestris*; Bdd, *B. diversus diversus*; Bu, *B. ussurensis*; and Bh, *B. honshuensis*. The number of samples is indicated in brankets. Error bars represent standard errors.

**Figure 5 insects-14-00340-f005:**
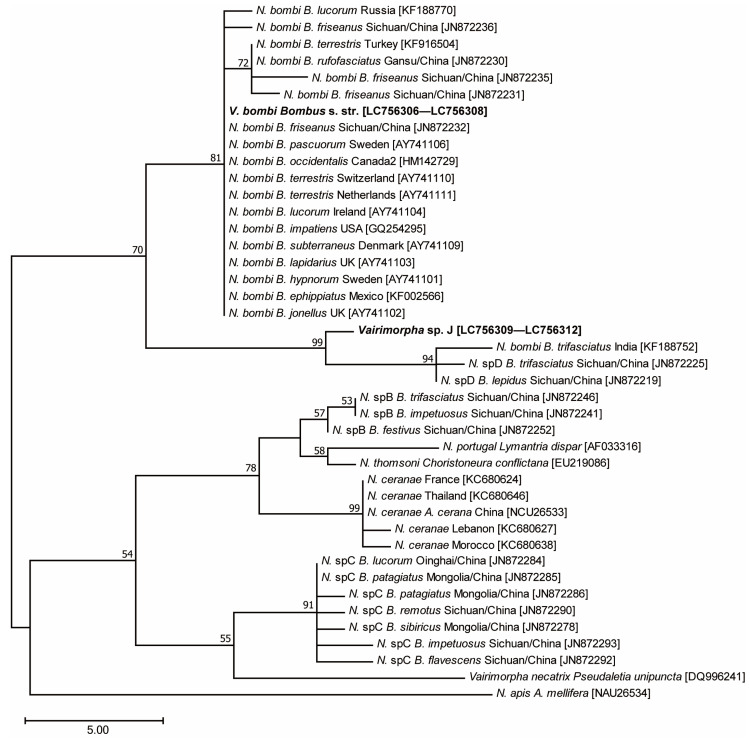
Maximum parsimony of the relationships among *Vairimorpha* spp. Samples collected from Hokkaido and Honshu based on alignments of the SSUr RNA gene. Bootstrap values exceeding 50% are shown (1000 replicates). The names indicate *Vairimorpha* spp., host names, collection site, and GenBank accession numbers were in square brackets.

**Table 1 insects-14-00340-t001:** List of bumblebee species collected in Biei and Kamifurano, Hokkaido and results of the PCR amplification of small subunit (SSU) rRNA and microscopy.

Year	Species *					
	Bt	Bha	Bhk	Bp	Bdt	Total Specimen Number
2014	Q79 (1/0) W30 (0/0) M19 (0/0)	Q106 (11/9) W56 (7/0) M33 (2/1)	Q1 (0/0)	W11 (0/0) M1 (0/0)	W1 (1/0)	337 (22/10)
2016	Q61 (1/0) W9 (0/0)	Q65 (8/0) W2 (0/0)	Q1 (0/0) W10 (0/0)			148 (9/0)
2018	W42 (1/0)	W45 (3/2)				87 (4/2)
Total number	240 (3/0)	307 (31/12)	12 (0/0)	12 (0/0)	1 (1/0)	572 (35/12)

* Species names and castes as abbreviations: Bt, *B. terrestris*; Bhs, *B. hypocrita sapporoensis*; Bhk, *B. hypnorum koropokkrus*; Bp, *B. pseudobaicalensis*; Bdt, *B. diversus tersatus*; Q, queen; W, worker; and M: male.

**Table 2 insects-14-00340-t002:** List of bumblebee species collected in Honshu and results of the PCR amplification of small subunit (SSU) rRNA and microscopy.

Year	Species *							Total Number
	Bhh	Bi	Baa	Bh	Bdd	Bu	Bc	
**Hinohara, Tokyo**								
2016	Q1 (0/0) W1 (0/0)		Q1 (0/0) W23 (0/0)		Q7 (7/0)			33 (7/0)
**Oshino, Yamanashi**								
2016	Q10 (0/0)		Q10 (0/0)		Q1 (1/0)			21 (1/0)
**east OMA**								
2016	Q4 (4/4) W11 (4/1) M5 (4/2)			W27 (4/0) M13 (0/0)	W14 (14/0)		W2 (0/0)	62 (30/7)
2019	Q18 W27 M4 (-/15) **							49 (-/15)
2020	Q2 (0/0) W18 (1/0) M3 (0/0)							23 (0/1)
**west OMA**								
2014		W8 (1/0) M7 (0/0)		M1(0/0)	W17(17/0) M5(5/0)			38 (23/0)
2016	W5 (0/0) M6 (1/0)	W46 (0/0)		W5(0/0)	W68(65/0)	W6(6/0)		136 (72/0)
**Hara, Nagano**								
2016	M2 (0/0)	W3 (0/0)		W3 (0/0) M1 (0/0)	W9 (7/0)			18 (7/0)
Total number	117 (14+/24)	64 (1/0)	34 (0/0)	50 (4/0)	121 (116/0)	6 (6/0)	2 (0/0)	380 (140+/23)

* Species names and castes as abbreviations: Bhh, *B. hypocrita hypocrita*; Bi, *B. ignitus*; Baa, *B. ardens ardens*; Bh, *B. honshuensis*; Bdd, *B. diversus diversus*; Bu, *B. ussurensis*; and Bc, *B. consobrinus;* Q, queen; W, worker; and M: male. ** PCR was not conducted, and sex/caste was not distinguished.

## Data Availability

The datasets analyzed during the current study are available from the corresponding author upon reasonable request.

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
