# Peer review of "Infection Prevalence of Microsporidia Vairimorpha (Nosema) spp. in Japanese Bumblebees"

_insects, 2023, doi:10.3390/insects14040340_

Round 1

Reviewer 1 Report

The manuscript is in the scope of the journal. I have several comments which are included in the atteched manuscript. Overall the manuscript can be submitted after changes.

Author Response

We would like to thank you for thoroughly evaluating our manuscript. We revised the manuscripts to address the comments. Please see the attachment.

Author Response

This manuscript describes a survey of Vairimorpha spp in Japanese bumblebees, including a new species of Vairimorpha. The information is valuable, but the methods and manuscript contain significant errors. I will be necessary to correct these before publication. A good reference for all who study these pathogens is Weiss, L. M. and J. J. Becnel, eds. 2014. Microsporidia: Pathogens of opportunity. Wiley Blackwell, Ames IA, USA.

Response: We would like to thank you for thoroughly evaluating our manuscript. We added the reference of the book and revised the manuscripts to address the comments below.

Line 14 Microsporidia do have negative effects on their hosts, as the authors state in their manuscript.

Response: We have changed to “Microsporidia are intracellular parasites that have negative effects on their hosts of various invertebrate and vertebrate animals”. (L18-19)

Line 15 Many species of microsporidia a pathogenic to bees, including honey bees. Perhaps the authors are referring to bumble bees.

Response: We referred ”three species of microsporidia” to Vairimorpha apis, V. ceranae. V. bombi pathogenic to honeybee and bumblebees because according to the book by Weiss and Bacnel, “only two microsporidian parasites have been characterized that are known to be infective to honeybees: Nosema apis and Nosema ceranae”. Therefore, we have changed “bees” to “honeybees and bumblebees”. (L20)

Lines 31, 64 and elsewhere Microsporidia infect many species of animals, not just insects.

Response: We have changed “insects” to “invertebrate and vertebrate animals” (L34-35; L63)

Line 196 The wash of body parts with PBS was likely not effective in removing all spores. All bee

species are covered with branched hairs, which retain spores, pollen and other particles. The authors do not explain the presence of Vairimorpha in/on the head, legs, thorax. In general, these microsporidia infect tissue within the abdomen, not legs and heads. Probably the authors detected RNA in spores retained by these branched hairs.

Response: We appreciate your suggestion. It is possible to detect the RNA of spores in hairs. Therefore, we added the sentence in Discussion as follows: “It is possible that spores contaminated on the surface of the body were detected, but considering differences in spore density between these two species, this might be a species-specific difference in infections for target organs.” (L338-340)

Line 445 Why would Vairimorpha sp. J “confer a benefit to its host”? The authors give no reason

to suggest this.

Response: If they establish a mutualistic relationship, Vairimorpha sp. J can spread throughout a host population. Therefore, we have changed this sentence as follows: “In this scenario, it may confer a fitness benefit to its host by establishing a mutualistic relationship to spread throughout a host population”. (L362-364)

The paper by Tokarev et al https://doi.org/10.1016/j.jip.2019.107279 should be cited, because it

provides a rationale for redefining the genera Nosema and Vairimorpha. This will be helpful to the readers.

Response: We have added this reference and changed to “Vairimorpha (Nosema) bombi, redefined by Tokarev et al. [7]”. (L67-68)

Author Response

The manuscript describes the determination of the prevalence of Vairimorpha (Nosema) bombi in various species and subspecies of bumblebees sampled at 4 sites in Japan. In the manuscript, it is disturbing that the results given in the text really contain a lot of data that are difficult to understand accurately. It would be necessary to graphically present as many of these results as possible. Tables  2 and S3 (S1 is missing) are also listed but are not attached to the manuscript. It is necessary to reanalyzing the results and presenting the data more graphically to make the manuscript more understandable. The 'new' Vairimorpha sp. J is mentioned mainly in the discussion. It would definitely be necessary to write more about the detection of a new pathogen or to present this topic in another article.

Response: We would like to thank you for thoroughly evaluating our manuscript. We added the reference of the book and revised the manuscripts to address the comments below.

Some comments:

- Row 15: the word ‘are’ is written twice

Response: We have deleted “are” before “and” (L20).

- Row 20 - 23: this sentence is written unclearly

Response: We have changed to “In this study, we investigated the infection of V. bombi among native Japanese bumblebees and the alien species, Bombus terrestris, established in Japan, using PCR and microscopy”. (L22-24)

- Row 64-68: not relevant to this article

Response: We have changed the first sentence of the paragraph as follows: “Microsporidia are intracellular spore-forming parasites of a variety of invertebrates and vertebrates and sometimes cause extensive damage and often death of the host [4]” (L62-64), and then these two example of this explanation.

- Row 68: In the article, you use the name Vairimorpha bombi for the pathogen commonly known as Nosema bombi. Since this is a relatively new name that is not yet widely accepted and known, add a description of the name change to the introduction and cite relevant literature.

Response: We have added the Tokarev et al. as follows: “Vairimorpha (Nosema) bombi, redefined by Tokarev et al. [7]”. (L67-68)

- Row 73 - 75: In the sentence you cite studies from the last decade, but most of the literature cited is older than 10 years. Since the number of such studies has increased greatly in recent years, find and cite the most recent literature on the topic. Also briefly describe the results of the cited studies.

Response: We added three more papers recently published and described the results as follows: “During the past decade, surveys about V. bombi infections among bumblebees have been conducted worldwide; their prevalence varied, ranging from 1.2% to 66.5% [2,11–24]” in Introduction (L72-74) and “In its native ranges, B. terrestris was infected by V. bombi with a relatively high prevalence at approximately 25%, sometimes more than 50% [13, 14, 24]. Since introduced for pollination in Chile, B. terrestris also has been established in Argentina’s Patagonia region [45] and was highly infected by V. bombi (37%), suggesting a potential threat to Argentina’s native bumblebees [46].” (L99-104)

- Row 125: perhaps the samples were treated individually?

Response: We have changed to “The midgut of each specimen was removed and crushed”. (L121-1222)

- Row 131: The table is very confusing. Edit the data to fit the available space (smaller font, etc.). You can replace the sampling sites in the table with the markers Site 1, Site 2... and add the description in the legend. In the legend, explain all the symbols used in the table (e.g. Q, W, M), not just the abbreviations of the bumblebee species. I assume that in ( ) is the number of positive results - this does not belong in the Materials and Methods chapter, but in the Results. If such an entry remains, you must explain it in the table heading or legend.

Response: We have separated the tables for Hokkaido and Honshu and deleted some words. Furthermore, we added an explanation about symbols (Q, W, M). I am not sure why this table was located in Material and Methods despite of cited in Results. I will suggest the editor to locate the table in Results.

- Row 137 -141: this method is quite incomparable between individual samples. Counting the spores in a hemocytometer (counting chamber) would be more accurate.

Response: We appreciate your suggestion. However, most of the specimens were not infected by V. bombi, so we used this method to determine if the specimens were infected by microscope.

- Row 142 - 190: add references of the methods used

Response: We cited the references for primers and MEGA (we mistook the spelling “Kumar et al. 2016”, so corrected and added the reference).

- Row 165: Where is table S2?

Response: We uploaded the supplemental tables as a different file.

- Row 232 - 243: It is impossible to say with certainty by microscopic examination alone that this is V. bombi, since the individual species of Vairimprpha (Nosema) spp. cannot be distinguished under the microscope. I suggest naming Vairimorpha spp. instead of V. bombi. This applies to all examples of microscopic examinations in the manuscript.

Response: As par your suggestion, we have changed the name to Vairimorpha spp. instead of V. bombi in Material & Method and Results including PCR examination because the primers also amplify Vairimorpha sp. J.

- Row 249-269: also present these results graphically to make them more understandable

Response: We have added the figure (Fig. 1) for the results.

- Row 302 -303: what do the labels (1), (2) and (3) mean?

Response: We categorized the specimens to analyze spore numbers. The labels (1), (2) and (3) mean the groups of them. We have changed this sentence as follows: “The number of Vairimorpha spp. spores in the midguts per 100 ng DNA containing the host gene was estimated using quantitative PCR in the following specimens: (1) sporulating Vairimorpha spp. PCR-positive bees, (2) non/low sporulating Vairimorpha spp. PCR-positive bees, and (3) non/low sporulating Vairimorpha spp. PCR-negative bees”. (L255-259)

- Row 303: Where is table S3?

Response: We uploaded the supplemental tables as a different file.

- Row 306, 308: ‘with more/less than five spores’’ /per visual field of the microscope?

Response: We have added “/per visual field of the microscope”. (L261, 263)

- Row 318: ‘t’ ?

Response: We have changed “t” to “the”. (Fig. 4)

- Figure 3: the vertical axis of the chart: Spore number per (not ‘par’) 100 ng DNA

Response: We have changed to “per”.

- Row 352: The PCR method is also specific, which microscopic examination is not, add 'and

specific'?

Response: We have added “and specific” before method. (L287)

- Row 414-416: where is this result listed in the results section?

Response: We have changed this sentence as par the other reviewer’s comment as follows: “Organ specificity in bumblebees differed between V. bombi and Vairimorpha sp. J: V. bombi showed systemic infectivity, whereas Vairimorpha sp. J did not seem to infect the head tissue of bees.” Then we have added “(Fig. 2)” at the end of the sentence. (L338)

Round 2

Reviewer 2 Report

The book by Weiss and Becnel is very good, but there have been more recent developments.  Please see the article by Chermurot et al (2017) European J of Protistology 61:13-19 on the third species of MIcrosporidia in honey bees.

Author Response

The book by Weiss and Becnel is very good, but there have been more recent developments. Please see the article by Chermurot et al (2017) European J of Protistology 61:13-19 on the third species of MIcrosporidia in honey bees.

Response: We would like to thank you for your advice. Since the number of microsporidia species is mentioned only in Summary, we deleted it and changed as follows: “The microsporidia pathogenic to honeybees and bumblebees belong to the genus Vairimorpha (previously Nosema)” .